# Current Advances in Basic and Translational Research of Cholangiocarcinoma

**DOI:** 10.3390/cancers13133307

**Published:** 2021-07-01

**Authors:** Keisaku Sato, Leonardo Baiocchi, Lindsey Kennedy, Wenjun Zhang, Burcin Ekser, Shannon Glaser, Heather Francis, Gianfranco Alpini

**Affiliations:** 1Division of Gastroenterology and Hepatology, Department of Medicine, Indiana University School of Medicine, Indianapolis, IN 46202, USA; linkenn@iu.edu (L.K.); heafranc@iu.edu (H.F.); galpini@iu.edu (G.A.); 2Hepatology Unit, Department of Medicine, University of Tor Vergata, 00133 Rome, Italy; baiocchi@uniroma2.it; 3Department of Research, Richard L. Roudebush VA Medical Center, Indianapolis, IN 46202, USA; 4Division of Transplant Surgery, Department of Surgery, Indiana University School of Medicine, Indianapolis, IN 46202, USA; wenzhang@iu.edu (W.Z.); bekser@iupui.edu (B.E.); 5Department of Medical Physiology, Texas A&M University College of Medicine, Bryan, TX 77807, USA; sglaser@tamu.edu

**Keywords:** cholangiocarcinoma, genetic aberrations, biomarkers, patient-derived xenograft, organoids

## Abstract

**Simple Summary:**

Cholangiocarcinoma (CCA) is highly malignant biliary tract cancer, which is characterized by limited treatment options and poor prognosis. Basic science studies to seek therapies for CCA are also limited due to lack of gold-standard experimental models and heterogeneity of CCA resulting in various genetic alterations and origins of tumor cells. Recent studies have developed new experimental models and techniques that may facilitate CCA studies leading to the development of novel treatments. This review summarizes the update in current basic studies of CCA.

**Abstract:**

Cholangiocarcinoma (CCA) is a type of biliary tract cancer emerging from the biliary tree. CCA is the second most common primary liver cancer after hepatocellular carcinoma and is highly aggressive resulting in poor prognosis and patient survival. Treatment options for CCA patients are limited since early diagnosis is challenging, and the efficacy of chemotherapy or radiotherapy is also limited because CCA is a heterogeneous malignancy. Basic research is important for CCA to establish novel diagnostic testing and more effective therapies. Previous studies have introduced new techniques and methodologies for animal models, in vitro models, and biomarkers. Recent experimental strategies include patient-derived xenograft, syngeneic mouse models, and CCA organoids to mimic heterogeneous CCA characteristics of each patient or three-dimensional cellular architecture in vitro. Recent studies have identified various novel CCA biomarkers, especially non-coding RNAs that were associated with poor prognosis or metastases in CCA patients. This review summarizes current advances and limitations in basic and translational studies of CCA.

## 1. Introduction

Cholangiocarcinoma (CCA) emerges from the biliary epithelium and is the second most common primary liver cancer after hepatocellular carcinoma (HCC) [1]. CCA is a relatively rare cancer, but worldwide incidence has been increasing in past years [2]. CCA is an aggressive malignancy, and early diagnosis is typically challenging due to asymptomatic characteristics at the early stage, resulting in poor prognosis with five-year survival under 20% [1,3]. Thus, treatment options are limited for diagnosed patients, who, in the majority of the cases, are already at advanced stages. Although liver transplantation or curative surgical resection is the sole effective procedures, CCA patients often have recurrence after surgery. A meta-analysis for 428 patients with unresectable CCA showed high recurrence rates after three years of orthotopic liver transplantation for patients with (24.1%) or without (51.7%) neoadjuvant chemoradiation [4]. Therefore, there is a critical need to develop novel diagnostic testing and therapeutic strategies for CCA [5].

Unique characteristics of CCA provide difficulties in basic and translational studies and prevent research progress in the field. The major barriers in CCA studies come from its heterogeneity. CCA is categorized into three types depending on the anatomical location of the tumor: intrahepatic (iCCA), perihilar (pCCA), and distal (dCCA) [5,6]. Since CCA tumor cells have biliary phenotypes, such as cytokeratin (CK)-7 and CK-19 expression [7], cholangiocytes have been established as the primary source of CCA tumors. Primary sclerosing cholangitis (PSC) is a bile duct disorder characterized by ductular reaction, biliary damage and inflammation, and liver fibrosis, and patients with PSC are at increased risk for CCA development, indicating carcinogenesis in cholangiocytes [8]; however, CCA can emerge from hepatic progenitor cells (HPCs) localized in the Canals of Hering, and this type of CCA is referred to as cholangiolocellular carcinoma [9,10]. Animal studies have shown that hepatocytes can initiate iCCA tumors with biliary phenotypes including CK-19 expression [11,12]. Furthermore, previous studies have reported combined hepatocellular cholangiocarcinoma (cHCC-CCA), which is liver cancer showing pathological features of both HCC and CCA [13,14,15]. Although the origin of cHCC-CCA may be HPCs, the pathophysiology of cHCC-CCA is largely undefined [16]. During liver injury, hepatocytes can transdifferentiate into cholangiocytes and vice versa, and HPCs can differentiate into both hepatocytes and cholangiocytes [17]. It is challenging to distinguish and identify each type of CCA and its origin. Pathophysiological characteristics of CCA tumors may differ depending on the type or origin of the tumor. This plasticity of hepatic cells and multiple potential origins of tumors contribute to the high heterogeneity of CCA and add to the complexity and difficulties in CCA research [18]. Recent studies have introduced novel biomarkers for diagnosis, experimental models, and advanced techniques to facilitate CCA studies. This review summarizes current advances and limitations in basic and translational research of CCA.

## 2. Current Advances in Basic CCA Research

### 2.1. Whole-Genome Screening

It is known that CCA tumors often have genetic mutations or aberrations. Previous studies identified various genetic mutations, and commonly mutated genes include *TP53*, *KRAS*, *IDH1/2*, and FGFR fusion [19,20]. Genomic profiling for 803 patients with biliary tract cancer found the most frequently altered genes, such as *TP53* (53%) and *KRAS* (26%), which were associated with poor prognosis in this cohort [21]. Based on these findings, clinical trials for IDH or FGFR inhibitors are ongoing, although results are often disappointing for CCA patients [5,22]. Since CCA is highly heterogeneous, only a limited percentage of patients have mutations or aberrations on targeted genes (under 10% for IDH1/2 mutations and 3–50% for FGFR2 fusion [22]), meaning the limited efficacy of drugs targeting mutated genes. Genome-wide screening to identify novel candidate genes that are altered in CCA tumors is ongoing research. A previous study analyzed 412 samples of biliary tract cancer including iCCA, pCCA, and dCCA by whole-exome sequencing and whole-genome sequencing [23]. This study identified commonly mutated genes, such as *TP53*, *KRAS*, and *SMAD4*, and found a novel deletion of *MUC17* on 7q22.1 [23]. Patients with *MUC17* deletion showed poor survival rates [23]. The landscape of genome and transcriptome in CCA may facilitate identification of target genes for novel therapies. Whole-exome and transcriptome sequencing for tumor and corresponding peritumor tissue samples of 9 iCCA patients identified an average of 378 somatic single nucleotide variants and 2366 differentially expressed genes in tumor tissues [24]. Interaction networks have shown that somatic mutations are highly correlated with altered gene expression, and mutations in key genes, such as *TP53*, may alter expression of numbers of genes [24]. Whole-exome sequencing for 318 iCCA patients identified 32 mutated genes associated with poor survival rates, which included *TP53* and *KRAS* [25]. Genome-wide transcriptomic profiling for iCCA tumors developed a transcriptomic panel with 8 genes, *BIRC5*, *CDC20*, *CDH2*, *CENPW*, *JPH1*, *MAD2L1*, *NEIL3*, and *POC1A*, which were robustly identified in patients with recurrence [26]. This study has demonstrated that the combination of this transcriptomic panel and clinical features, such as tumor size, could be useful to predict the risk of CCA recurrence [26]. A previous study categorized 133 cHCC-CCA samples into three subtypes according to the localization of HCC and CCA tumors: separate, combined, and mixed [27]. Genomic and transcriptomic sequencing when comparing HCC and iCCA found that genomic landscape and transcriptomic profiles of the combined type of cHCC-CCA was similar to those of iCCA, and the mixed type of cHCC-CCA was similar to HCC, indicating that therapies for cHCC-CCA may be adopted depending on subtypes [27]. This study showed that a high percentage of cHCC-CCA tumors expressed Nestin compared to HCC or iCCA, and positivity of Nestin expression was associated with poor survival rates, suggesting that Nestin could be useful as a novel biomarker for diagnosis testing of cHCC-CCA [27].

DNA methylation regulates gene expression, and hypermethylation on specific promoter CpG islands have been identified in CCA tumors, which may play a key role in the pathophysiology of CCA [20]. Genome-wide analysis for promoter methylation identified hypermethylation on promoter regions of genes associated with Wnt signaling, such as SFRP2, which expression levels were downregulated in CCA tumors [28]. Analysis of gene expression and gene methylation for CCA samples using datasets obtained from Gene Expression Omnibus (GEO) database found 98 hypermethylated, downregulated genes and 93 hypomethylated, upregulated genes [29]. Another analysis using GEO datasets showed that CCA patients with widespread hypermethylation of promoter-related CpG sites suffered unfavorable prognosis with poor survival rates [30]. Genome-wide profiling of DNA methylation and gene/microRNA (miRNA) expression using data obtained from The Cancer Genome Atlas (TCGA) have shown altered DNA methylation on 12,259 CpGs and altered expression patterns in 3,305 genes and 101 miRNAs [31]. This study identified candidate genes and miRNAs, such as MDK, DEPDC1, miR-22, and miR-551b, which could be useful as prognostic biomarkers of CCA [31]. These studies show that whole-genome screening and landscape of mutations, genome, transcriptome, and methylation may lead to identification of novel biomarkers and therapeutic targets of CCA; however, a genome-wide association study for CCA is still not available to date, which may provide novel understandings of genetic variations associated with CCA.

### 2.2. Animal Models

Availability of experimental models is a key factor to facilitate basic and translational research; however, no gold standard CCA models are available to date regardless of various animal models introduced in previous studies [32,33,34]. Xenograft models generated by subcutaneous transplantation of human CCA cell lines into flanks of nude mice are still one of the most commonly used models in CCA studies, but this model does not mimic CCA tumorigenesis and microenvironment in addition to the species mismatched conditions [33]. Administration of toxins, such as thioacetamide (TAA) and diethylnitrosamine (DEN), was used in previous studies to generate CCA tumors in rodents; however, ~22-week TAA administration for rats [35] and combination of DEN treatments with bile duct ligation (BDL)-induced biliary damage in mice [36] do not mimic pathological conditions of CCA patients. CCA tumors often have mutations in specific genes, such as *TP53* and *KRAS*, and studies in past years developed various genetically engineered mouse models targeting these genes [32,33,34]. Mice with a mutation in a target gene, such as Kras^G12D^ mice, which have *Alb*-Cre-mediated somatic KRAS activation, showed iCCA-like tumor in vivo, although tumors showed low penetrance and long latency [37]. Kras^G12D^;p53^L/L^ mice, which have liver-specific KRAS activation and p53 deletion, showed higher penetrance and iCCA tumors in a short time [37]. The chance of CCA development in a practically reasonable time frame is limited in mice with a mutation or deletion in a single target gene; therefore, to increase penetrance and shorten the period of CCA tumor development, two or more genetic mutations or deletions are increasingly introduced in current CCA mouse models [32,33,34]. However, CCA is heterogeneous and only a limited percentage of CCA patients share mutations in specific genes. Conditions in these genetically engineered models (e.g., *Alb*-Cre-mediated Smad4^Co/Co^Pten^Co/Co^ mice [38]) are rare in CCA patients. In addition, some models, such as *Alb*-Cre-mediated Pten^f/f^Grp94^f/f^ mice [39], show both HCC and CCA. It is unclear if these mice mimic conditions of human patients with CCA or cHCC-CCA. Since *Alb*-Cre mice have Cre expression in hepatocytes and hepatic stellate cells [40], not cholangiocytes, these mice may have high possibilities to have a mixture of HCC and CCA. Searching for ideal CCA models still continues in current studies. Injection of three oncogenic plasmids, myristoylated AKT1, mutated YAP, and pCMV-Sleeping Beauty, into mouse livers generated iCCA-like peripheral tumors and pCCA-like perihilar tumors after 10 weeks [41]. Gene and protein expression levels, such as *Fgfr2* and αSMA, were different between peripheral and perihilar tumors, showing that characteristics of tumors generated by the same technique can differ depending on the tumor location [41]. A previous study has demonstrated that *Opn*-Cre^ER^ triggers recombination in 99.9% of cholangiocytes while *Ck19*-Cre^ER^ has only 32% recombination efficiency [42]. A subsequent study generated *Opn*-Cre-mediated cholangiocyte-specific Kras^G12D^ mice and fed them with 3,5-diethoxycarbonyl-1,4-dihydrocollidine (DDC) diet [43]. DDC diet induces bile duct damage and ductular reaction in mice [17]. DDC-fed Kras^G12D^ mice showed iCCA tumors after 21 weeks [43]. This study has demonstrated that iCCA tumors in this model, as well as human patients overexpress Tensin-4, which drives CCA cell proliferation [43]. *Alb*-Cre-mediated Kras^G12D^/CDH1^ΔL^ mice showed tumors after 8 months of age, but most of tumors were HCC, and CCA was only 10% in the established tumors [44]. Feeding with high-fat diet generated liver tumors in Kras^G12D^/CDH1^ΔL^ mice at 5 months of age, and 34% of tumors was CCA, showing that high-fat diet facilitates tumorigenesis in the liver and increases the possibility of CCA development [45]. These studies indicate that the combination of genetic alterations and feeding diets may significantly affect penetrance and the time period of cancer development in the liver as well as phenotypes of cancer (HCC, CCA, or mix). Table 1 lists selected genetically engineered CCA mouse models used in current CCA studies. Limitations of these models include the lack of tumorigenic conditions in patients with cholangiopathies. As mentioned, PSC is a common risk factor for CCA development; however, the most common PSC model mice, *Mdr2*^−/−^ mice show only HCC, not CCA, at 7–12 months of age regardless of their genetic backgrounds [46]. Detailed mechanisms of biliary tumorigenesis in cholangiopathies are undefined, and appropriate animal models resembling conditions in patients are still unavailable.

Although it is still common to generate human CCA tumors in mice by transplantation of CCA cell lines, such as Mz-ChA-1 cells [33], recent studies have introduced techniques to generate CCA tumors using tissues obtained from patients. A previous study subcutaneously transplanted fresh iCCA tumors excised from patients into NOD/SCID mice [47]. Although only one tumor out of 17 (5.8%) was successfully engrafted after 4 months, generated tumors could be explanted and implanted into new mice [47]. This patient-derived xenograft (PDX) mouse model maintained histological tumor features as well as genetic alterations, such as KRAS G12D mutation, with the original tumor [47]. Another study generated PDX mice highly successfully (75%) by transplantation of frozen CCA tissues into Balb/c *Rag-2*^−/−^/*Jak3*^−/−^ mice [48]. This study established four novel CCA cell lines from generated PDX mice, and those CCA cell lines showed high transplantation efficiency (up to 100%), which may be useful for generation of xenograft CCA models [48]. A large scale study used tumor samples created from surgical resection specimens or radiographic biopsies of 87 patients with biliary tract cancers including iCCA, pCCA, and dCCA [49]. Out of 87 patient specimens, 47 PDX models were successfully generated in NOD/SCID mice [49]. Generated PDX mice maintained histological and genetic characteristics compared to original tumors in patients [49]. This study has demonstrated that patients whose specimen are successful for engraftment suffer poor survival rates compared to patients with tumors that do not generate PDX mice [49]. PDX mouse models resemble unique and heterogeneous characteristics of CCA tumors in the individual patient. These models may be useful for biomarker detection, mutation screening, and drug testing, which may lead to customized treatments for each patient.

### 2.3. In Vitro Models

Various human CCA cell lines, such as HuCCT1 and TFK-1, have been used for in vitro studies, and genomic and transcriptomic analysis showed that these cell lines shared similar mutational signatures and transcriptomic profiles compared to primary tumors [50]. Due to the heterogeneity of CCA, multiple cell lines are usually used to confirm results, and cell line authentication by detecting short tandem repeats is required to determine genetic stability and contamination. Novel human CCA cell lines have been established in recent studies from PDX mice [48] or iCCA and pCCA tumor tissues [51]. Recent studies have established the methodology to generate and characterize organoids, three-dimensional (3D) cultured mini-organs, using primary liver tissues of patients with cholangiopathies or CCA [52,53]. A previous study established CCA organoids with 50% success rates using resected tissue specimens of iCCA patients and maintained them over one year [54]. Established organoids showed similar histopathological features to primary tumors [54]. Organoids mimic 3D cellular architecture and interaction more appropriately compared to two-dimensional (2D) monolayer cultures and may be suitable for in vitro models in CCA studies [52]. A previous study cultured patient-derived iCCA cells in 2D or 3D culture systems and compared cellular functions and expressions [55]. CCA organoids showed significantly higher liver function indexes, such as expression levels of alanine aminotransferase (ALT) and aspartate aminotransferase (AST), as well as elevated fibrosis indexes including matrix metalloproteinase (MMP) expression compared to same CCA cells cultured as 2D monolayers, indicating that cell functions and characteristics may differ between 2D and 3D culture systems, and CCA cells may be more functional in 3D cultures [55]. CCA organoids are useful for drug testing or genetic screening. A previous study cultured CCA organoids established from iCCA tissues in glucose-free media and found that CCA organoids showed reduced proliferation but elevated gemcitabine resistance in glucose-free media [56]. Patient-derived CCA organoids showed enlarged mitochondria compared to organoids generated from normal liver tissues, and knockdown of genes that are associated with mitochondrial fusion process, *OPA1* and *MFN1*, inhibited mitochondrial fusion and cell viability in populating cells of CCA organoids [57]. A study using these CCA and normal liver organoids found that CCA organoids expressed elevated lysyl-tRNA synthetase (KARS), and treatment of KARS inhibitor cladosporin decreased cell viability of CCA organoid cells, showing potential anti-cancer effects of cladosporin for CCA [58]. 

A recent study isolated cholangiocytes from bile ducts of wild-type (WT) or *Ink4a/Arf*^−/−^ mice and transfected them with retrovirus vectors for KRAS^G12V^ to generate malignant murine cholangiocytes [59]. Although KRAS^G12V^ expression inhibited proliferation of WT cholangiocytes, transplantation of KRAS^G12V^-expressing *Ink4a/Arf*^−/−^ cholangiocytes generated CCA tumors in WT mice, and CCA organoids were established from tumor tissues of these syngeneic CCA mice [59]. This study has demonstrated that CCA organoids have cancer stem cell-like characteristics, such as high expression of *Cd24*, *Cd44*, and *Sca1*, as well as the ability to generate secondary tumors when transplanted in mice; however, the same cells cultured in 2D monolayers have decreased successful rates to generate CCA tumors after transplantation into syngeneic mice, indicating that organoids may be a better technique to retain CCA cell functions in vitro compared to classic monolayer cultures [59].

Another study established hepatic organoids using liver tissues explanted from WT or *TP53*^−/−^ mice, and FGFR2 fusion proteins identified in CCA patients, such as FGFR2-TACC3 and FGFR2-BICC1, were expressed in established organoids by retroviruses [60]. Expression of FGFR2 fusion proteins inhibited WT organoid growth, but transplantation of *TP53*^−/−^ organoids expressing fusion proteins generated CCA tumors in NOD/SCID mice while *TP53*^−/−^ organoids with no fusion protein expression did not [60]. Although FGFR inhibitors may be therapeutic for CCA patients with FGFR2 fusion proteins, some patients show resistance and the efficacy of FGFR inhibitors may be limited [61]. This study has demonstrated that combination of FGFR kinase inhibitor BGJ398 and MEK1/2 inhibitor trametinib significantly inhibited tumor growth compared to treatments with BGJ398 alone using syngeneic CCA mouse models driven by FGFR2-BICC1-expressing organoids [60]. These studies showed that robust KRAS activation and FGFR2 fusion proteins inhibit proliferation and growth in normal hepatobiliary cells, and the combination with suppression/depletion of tumor-suppressor genes, such as *Ink4a/Arf* or *TP53*, is required for CCA tumorigenesis. Although syngeneic CCA mouse models were generated using murine organoids, patient-derived human organoids can also be the source of CCA tumors in xenograft mouse models. Transplantation of organoids established from human iCCA tumor tissues generated xenograft tumors in NOD/SCID mice, which show similar histological features compared to the original patient’s tumor [62]. Figure 1 summarizes recent experimental CCA models including PDX and organoid models, and Table 2 shows pros and cons for mentioned CCA animal models.

### 2.4. Biomarkers

Since early diagnosis is still challenging, searching for novel biomarkers that could be used for diagnostic testing or prediction of prognosis continues in current CCA studies. Various candidate biomarkers have been identified in previous studies, and the majority of current biomarkers includes genetic mutations, such as *TP53* and *KRAS* mutations, proteins including cytokines, and non-coding RNAs [63,64]. This review focuses on recent advances in biomarker search, and for more information of other currently known CCA biomarkers, see previous reviews [63,64]. Samples to be analyzed in previous studies are usually tissue samples or fluid samples, such as serum, bile, or urine. For example, analysis of RNA-sequencing (RNA-seq) data obtained from TCGA, which include 36 CCA tissues and 9 adjacent normal tissues, identified bloom syndrome helicase (BLM) as a potential CCA biomarker [65]. BLM is upregulated in CCA tumors and cell lines, and high BLM expression is associated with poor survival rates [65]. Receiver operating characteristic (ROC) curve analysis showed that BLM could be useful for diagnostic testing [65]. Another study analyzing RNA-seq data obtained from TCGA identified multiple candidate biomarkers, such as cyclin-dependent kinase 1 (CDK1), which is significantly upregulated in CCA tumors and associated with poor survival rates [66]. Immunohistochemistry for liver tissues identified overexpression of cadherin 17 (CDH17 or CA17) in CCA, which was associated with poor survival rates [67]. Serum levels of carbohydrate antigen 19-9 (CA 19-9) is clinically useful as a biomarker of CCA [63], and a previous study has demonstrated that the combination of serum levels of CA 19-9 and dickkopf-related protein 1 (DKK-1) could provide better diagnostic and prognostic performance compared to CA 19-9 alone in iCCA patients [68].

Extracellular vesicles (EVs) are small particles secreted from cells and contain cargo proteins, DNAs, and RNAs, which regulate cellular functions in recipient cells [69]. EV-mediated intercellular communication may play a vital role in the pathophysiology of cholangiopathies as well as CCA [70,71,72]; therefore, EVs secreted from CCA tumors may contain unique cargo mediators that could be used as early diagnostic biomarkers. EVs isolated from serum samples of patients with CCA contained elevated levels of proteins, including pantetheinase and C-reactive protein, compared to EVs isolated from healthy individuals or patients with PSC or HCC [73]. Analysis of miRNAs contained in serum EVs of 36 CCA patients and 12 healthy individuals identified miR-200 family enriched in CCA EVs, and these miRNAs showed higher area under the ROC curve (AUC) than CA 19-9 [74]. Proteomic analysis for EVs isolated from human bile samples of 10 CCA and 10 choledocholithiasis (bile duct stone) patients identified 166 proteins as CCA-specific [75]. This study has demonstrated that CCA EVs contain significantly higher levels of claudin-3, which could be a useful biomarker to distinguish CCA and bile duct stones [75]. Transcriptomic analysis for isolated EVs from serum or urine samples found that EVs from CCA patients contain robust levels of various mRNAs, such as *CMIP* for serum EVs and *UBE2C* for urine EVs, compared to EVs isolated from healthy individuals or patients with PSC or ulcerative colitis [76]. These studies show the potentials of EVs as biomarker carriers and important samples for diagnostic testing.

Recent advances in CCA biomarkers include identification of non-coding RNAs, such as miRNAs, long non-coding RNAs (lncRNAs), circular RNAs (circRNAs), and P-element-induced wimpy testis (PIWI)-interacting RNAs (piRNAs). Numbers of candidate non-coding RNAs have been identified as potential biomarkers from CCA tissues or cell lines. For example, whole-transcriptome sequencing for 8 CCA tumor and adjacent normal tissues detected 2,895 mRNAs, 56 miRNAs, 151 lncRNAs, and 110 circRNAs that were differentially expressed in CCA tumors [77]. It is known that miR-21 and miR-122 are associated with CCA, and combined validation of plasma levels of miR-21, miR-122, and CA 19-9 showed better AUC compared to CA 19-9, showing the potentials of non-coding RNAs as diagnostic biomarkers to distinguish iCCA and control individuals [78]. This review introduces recently identified non-coding RNAs, which are associated with poor prognosis or the pathophysiology of CCA. For more information of other non-coding RNAs, see previous review articles [79,80,81,82]. For miRNAs, a previous study analyzed TCGA dataset of CCA and found that expression levels of miR-3913 were elevated in CCA tumors [83]. High expression of miR-3913 was associated with poor survival rates in CCA patients, indicating the potential as a prognostic biomarker [83]. A study using 30 CCA tumor tissues and 20 adjacent normal tissues showed that miR-29b was significantly downregulated in CCA tumors, and low miR-29b expression was associated with poor survival rates [84]. This study has demonstrated that miR-29b targets and regulates expression of DNA methyltransferase 3 beta (DNMT3B), which is upregulated in CCA tumors and induces CCA growth, indicating the potential of miR-29b/DNMT3B as a therapeutic target [84]. Other recent studies have identified miR-150, miR-144, miR-451a, miR-1182, and let-7a as being associated with CCA and could be useful as biomarkers [85,86,87] (Table 3).

Numerous studies have identified various lncRNAs, which are associated with CCA [81]. lncRNAs have binding sites for target miRNAs and inhibit their functions by sponging. As a result, expression levels of genes, which are targeted by sponged miRNAs, will be upregulated due to lncRNAs. This function of lncRNAs as competing endogenous RNAs (ceRNAs) may be vital in the pathophysiology of CCA. Data analysis for TCGA dataset of 36 CCA tissues and 9 control tissues constructed ceRNA network associated with CCA [88]. CCA tumors may express unique lncRNAs regulating downstream miRNAs and gene expression, which could be useful as CCA biomarkers. Analysis of TCGA dataset identified five lncRNAs, which expression levels were altered in CCA tumors, and combination of these lncRNAs showed better AUC compared to single lncRNA and the potentials as a biomarker to predict poor prognosis and recurrence [89]. A previous study using CCA tumor and adjacent normal tissues obtained from 57 patients has demonstrated that lncRNA FOXD2-AS1 is upregulated in CCA tumors as well as CCA cell lines, and patients with high FOXD2-AS1 expression showed worse prognosis [90]. This study showed that FOXD2-AS1 inhibited miR-760 functions by sponging, resulting in upregulated expression of target gene of miR-760, oncogene E2 transcription factor 3 (E2F3) in CCA [90]. Inhibition of FOXD2-AS1 or E2F3 decreased CCA cell proliferation, but inhibition of miR-760 promoted proliferation in vitro [90]. Table 4 lists other lncRNAs identified in recent CCA studies [91,92,93,94,95,96,97].

circRNAs are a type of non-coding RNA that have a closed loop structure. Recent CCA studies revealed the association of circRNAs with CCA and that circRNAs could function as ceRNAs similar to lncRNAs, inhibiting target miRNA functions and inducing downstream gene expression. A previous study has demonstrated that the crcRNA circ-LAMP1 is upregulated in CCA tumor tissues and cell lines, and high expression of circ-LAMP1 is associated with poor survival rates [98]. This study showed that circ-LAMP had binding sites for miR-556 and miR-567, and both miRNAs regulated expression of YY1 [98]. Inhibition of circ-LAMP1 decreased cell proliferation and invasion of CCA cell lines in vitro, as well as CCA tumor growth in xenograft mouse models, indicating the potentials of circ-LAMP1 as a biomarker and therapeutic target of CCA [98]. The authors isolated EVs from bile samples of CCA patients and performed circRNA profiling and found that circ-CCAC1 was enriched in CCA bile EVs and was highly expressed in CCA tumors and cell lines, which was associated with poor prognosis in patients [99]. circ-CCAC1 sponges miR-514a, which regulates YY1 expression, and high levels of circ-CCAC1 in circulating EVs promoted CCA growth and metastases in xenograft mouse models [99]. A study using 35 paired CCA tissues and adjacent normal tissues identified elevated expression of circ-DNM3OS associated with TNM stage and lymph node invasion in CCA patients [100]. circ-DNM3OS induces MORC family CW-type zinc finger 2 (MORC2) expression via sponging miR-145, and inhibition of circ-DNM3OS decreased tumor growth by upregulation of miR-145 and downregulation of MORC2 in xenograft mice [100]. These studies demonstrated the pathophysiological roles of circRNAs as ceRNAs in CCA. Other circRNAs identified in recent CCA studies are listed in Table 5 [101,102,103,104,105].

piRNA is a class of small non-coding RNA, which makes an RNA-protein complex with PIWI proteins. Functions of piRNAs include post-transcriptional regulation of gene expression, which is similar to miRNAs, and piRNAs may be involved in cancer development and have the potentials as novel cancer biomarkers [106]. A previous study isolated EVs from serum samples of patients with gastric cancer, and found that cancer-derived serum EVs contained elevated levels of piRNAs, such as piR-019308, compared to EVs isolated from healthy individuals, and ROC curve analysis showed the diagnostic potentials of these piRNAs (for piR-019308, AUC = 0.82) [107]. Although studies are limited for CCA and functional roles of piRNAs in CCA are largely undefined, a previous study performed piRNA profiling for EVs isolated from plasma samples of patients with CCA or gallbladder carcinoma [108]. This study identified various piRNAs upregulated in CCA-derived EVs compared to normal EVs, such as piR-10506469, as well as downregulated piRNAs, such as miR-17802142 [108]. Cargo levels of piR-10506469 in plasma EVs of CCA patients were significantly decreased after one week of surgery, indicating the correlation of EV piR-10506469 levels with CCA tumors [108]. Future studies may identify novel piRNAs associated with CCA development leading to novel diagnostic testing.

## 3. Emerging Roles of Gut Microbiota in CCA

One of the emerging fields in CCA studies includes the functional roles of gut microbiota. As mentioned earlier, PSC is a common risk factor for CCA development [8]. Previous studies showed that PSC patients had different gut bacteria profiles with decreased diversity, which is associated with cholestatic liver injury [109]; therefore, gut microbiota may be associated with biliary carcinogenesis and CCA. A previous study analyzed gut microbiota using stool samples from patients with HCC, iCCA, or liver cirrhosis and healthy individuals and found that iCCA patients had abundance of *Lactobacillus*, *Actinomyces*, *Peptostreptococcaceae*, and *Alloscardovia* genera compared to other patient groups [110]. Fluke infection with *Opisthorchis viverrini* or *Clonorchis sinensis* is a common risk factor for CCA, and *O. viverrini* infection in hamsters is an animal model to mimic CCA development [5]. Analysis of colorectal feces of hamsters demonstrated that *O. viverrini* infection altered gut microbiota, increasing *Lachnospiraceae*, *Ruminococcaceae*, and *Lactobacillaceae*, and decreasing *Porphyromonadaceae*, *Erysipelotrichaceae*, and *Eubacteriaceae* [111]. Myeloid-derived suppressor cells (MDSCs) are immunosuppressive cells with an ability to inactivate T cells, and high MDSC population is associated with CCA [112,113]. A previous study has demonstrated that cholestatic liver injury induces bacterial leakage of gut bacteria via elevated intestinal permeability, which promotes MDSC accumulation in the liver leading to CCA development in mouse models in vivo [114]. Although these studies indicate the association of gut bacteria with CCA, it is still undefined which genera/species of bacteria are involved in the pathophysiology of CCA. Further studies are required to elucidate the detailed mechanisms of CCA development induced by gut bacteria.

## 4. Conclusions and Future Perspectives

Although CCA studies are still limited and further studies are required, current studies have developed novel methodologies and techniques to facilitate CCA research. Important advances in CCA studies include new CCA models. PDX mouse models maintain genetic and histological features of primary tumors found in the patient and are suitable for genetic screening, drug testing, or even design of customized treatments for the patient. Syngeneic CCA mouse models have overcome the mismatch of species between CCA tumors and the host organisms. Xenograft mouse models established by transplantation of human CCA cell lines usually do not generate the tumor microenvironment, which is often observed in CCA patients [33]. Syngeneic CCA mouse models established by murine malignant cholangiocytes showed the tumor microenvironment-like conditions in vivo [59]. The tumor microenvironment is a dense stroma, which may play a vital role in the pathophysiology of CCA and could be a promising therapeutic target [115], although it is challenging to resemble the tumor microenvironment in current CCA animal models. In a previous study, rat CCA models established using malignant rat cholangiocyte lines, BDEneu cells, showed liver conditions like CCA tumor microenvironment [116]. Syngeneic CCA models may be suitable as animal models for studies of CCA tumor microenvironment. Although PDX mouse models maintain characteristics of patients’ primary CCA tumors, it is undefined if PDX models also resemble CCA microenvironment that is found in the patient. Since both syngeneic rat and mouse models showed CCA tumor microenvironment-like stroma, matched species between CCA tumors and the host animals may be required to generate CCA microenvironment in animals. Syngeneic animal models established using genetically modified cancerous cholangiocytes or organoids still do not mimic CCA development associated with biliary damage, such as PSC. Combination of biliary damage with other factors may be required to form CCA tumors. For example, only KRAS activation in cholangiocytes inhibited cell proliferation and did not generate CCA tumors, and knockout of *Ink4a/Arf* was also required [59]. DDC diet generated iCCA tumors in *Opn*-Cre-mediated cholangiocyte-specific Kras^G12D^ mice after 21 weeks but control feeding did not [43]. High-fat diet feeding increased the chance of CCA development (34%) in Kras^G12D^/CDH1^ΔL^ mice compared to control diet (10%) [44]. These studies indicate that CCA development may be mediated by multiple factors, damage, or mutations in hepatobiliary cells. Future studies may establish CCA development models from PSC (e.g., *Mdr2*^−/−^ mice with other factors) to mimic transitional conditions from biliary inflammation and damage to CCA. Establishment of CCA organoids is another important advance in CCA research. Compared to 2D cell culture, 3D organoid system can retain CCA tumor functions, such as the ability to generate secondary tumors in mice after excision and re-transplantation [59]. Data using 3D organoids may be required for future CCA studies to confirm results and functions of CCA tumor cells. Searching for genetic mutations, aberrations, methylations, or biomarkers is still ongoing in current basic studies, but accumulation of these studies may lead to the development of novel diagnostic testing or therapies. According to PubMed, papers or entries found in a search with “cholangiocarcinoma” have been increasing in every year (2010: 607, 2015: 1092, and 2020: 1682 entries). CCA studies may develop better models and identify novel therapeutic approach to overcome this rare heterogeneous cancer soon.

## Figures and Tables

**Figure 1 cancers-13-03307-f001:**
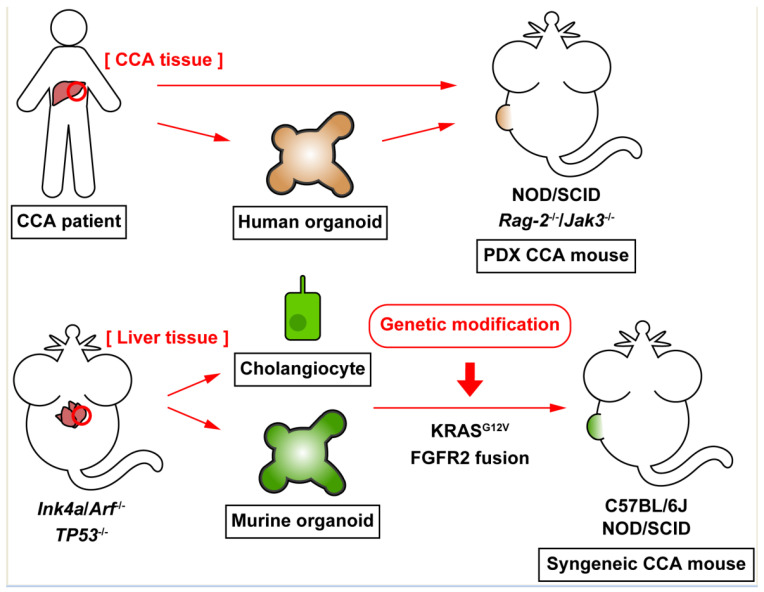
Current advances in experimental models of cholangiocarcinoma (CCA). CCA tumor tissues excised from patients can be engrafted in immunodeficient mice, such as NOD/SCID mice, to generate patient-derived xenograft (PDX) mouse models. CCA organoids, which are established from CCA tumor tissues, resemble 3D cellular architecture, and maintain functions of CCA tumor cells. Human CCA organoids can be transplanted into mice to generate xenograft animals. Murine cholangiocytes or hepatic organoids, which are derived from knockout mice for tumor-suppressor genes become malignant by genetic modification, such as KRAS activation or FGFR2 fusion protein expression, and these malignant cholangiocytes or organoids establish CCA tumors in WT or immunodeficient mice. These syngeneic CCA mouse models overcome the mismatch of species between CCA tumors and the host animals.

**Table 1 cancers-13-03307-t001:** Selected genetically engineered CCA mouse models.

Mouse Model	Recombination	Induced Alterations
Kras^G12D^ mice [37]	*Alb*-Cre	KRAS activation
Kras^G12D^;p53^L/L^ mice [37]	*Alb*-Cre	KRAS activation and p53 deletion
Smad4^Co/Co^Pten^Co/Co^ mice [38]	*Alb*-Cre	Deletion of SMAD4 and PTEN
Pten^f/f^Grp94^f/f^ mice [39]	*Alb*-Cre	Deletion of PTEN and GRP94
AKT/YAP Sleeping Beauty [41]	Sleeping Beauty transposon	Activation of AKT and YAP
Kras^G12D^ mice with DDC diet [43]	*Opn*-Cre	KRAS activation and biliary damage
Kras^G12D^/CDH1^ΔL^ mice with high fat diet [45]	*Alb*-Cre	KRAS activation, deletion of E-cadherin, and non-alcoholic fatty liver disease

**Table 2 cancers-13-03307-t002:** Brief characteristics of current CCA animal models.

Model	Pros	Cons
Carcinogen administration (TAA, DEN)	Established, reproducible, and easy procedures Allows to compare early stage and late stage	Long time administration to generate tumors Procedures established mainly for rats Does not mimic human conditions
Genetically engineered mouse	Mimics common genetic aberrations found in humans Allows to compare early stage and late stage	Double or triple knockout required to generate tumors Does not mimic CCA development associated with biliary damage and inflammation
Combination of genetically engineered mouse and special feeding	Only single or double knockout or mutation required Mimics biliary damage and inflammation by feeding	Limited previous studies Need to evaluate established tumors as CCA Relatively long period required to establish tumors
Xenograft mouse	Relatively easy procedures Established methodologies for mice Fast tumor formation	Mismatch species Lack of tumor microenvironment Cannot compare early stage and late stage
Patient-derived xenograft mouse	Maintains individual CCA characteristics Allows drug testing or genetic screening personalized for the donor patient	Mismatch species Need to maintain mice with tumors for each donor patient
Syngeneic CCA mouse	Matched species Mimics tumor microenvironment	Relatively challenging procedures Does not mimic CCA development associated with biliary damage and inflammation Cannot compare early stage and late stage

**Table 3 cancers-13-03307-t003:** Selected miRNAs identified in recent CCA studies.

miRNAs	Samples Analyzed	Expression in CCA	Targets	Association with Poor Survival
miR-22 [31]	CCA tumor	Downregulated	N/A	High expression
miR-551b [31]	CCA tumor	Downregulated	N/A	Low expression
miR-200 family [74]	Serum EV	Upregulated	N/A	High levels
miR-3913 [83]	CCA tumor	Upregulated	N/A	High expression
miR-29b [84]	CCA tumor	Downregulated	DNMT3B	Low expression
miR-150 [85]	Serum	Downregulated	N/A	N/A
miR-144 [86]	CCA tumor	Downregulated	ST8SIA4	N/A
miR-451a [86]	CCA tumor	Downregulated	ST8SIA4	N/A
miR-1182 [87]	CCA tumor	Downregulated	NUAK1	N/A
let-7a [87]	CCA tumor	Downregulated	NUAK1	N/A

**Table 4 cancers-13-03307-t004:** Selected lncRNAs identified in recent CCA studies.

lncRNAs	Samples Analyzed	Expression in CCA	Primary Targets	Secondary Targets
FOXD2-AS1 [90]	CCA tissues and cells	Upregulated	miR-760	E2F3
GAS5 [91]	CCA tissues and cells	Upregulated	miR-1297	N/A
TTN-AS1 [92]	CCA tissues and cells	Upregulated	miR-320a	NRP-1
PAICC [93]	CCA tissues and cells	Upregulated	miR-141-3p, miR-27a-3p	YAP1
SNHG16 [94]	CCA tissues and cells	Upregulated	miR-146a	GATA6
MT1JP [95]	CCA tissues and cells	Downregulated	miR-18a	FBP1
CASC2 [96]	CCA tissues and cells	Downregulated	miR-18a	SOCS5
HOTAIR [97]	CCA tissues and cells	Upregulated	miR-204	HMGB1

**Table 5 cancers-13-03307-t005:** Selected circRNAs identified in recent CCA studies.

circRNA	Samples Analyzed	Expression in CCA	Primary Targets	Secondary Targets
circ-LAMP1 [98]	CCA tissues and cells	Upregulated	miR-556, miR-567	YY1
circ-CCAC1 [99]	CCA tissues and cells, bile EVs	Upregulated	miR-514a	YY1
circ-DNM3OS [100]	CCA tissues and cells	Upregulated	miR-145	MORC2
circ-HIPK3 [101]	CCA tissues and cells	Upregulated	miR-148a-3p	ULK1
circ-0000284 [102]	CCA tissues and cells	Upregulated	miR-637	LY6E
circ-0005230 [103]	CCA tissues and cells	Upregulated	miR-1238, miR-1299	N/A
circ-SMARCA5 [104]	CCA tissues	Downregulated	N/A	N/A
circ-0000673 [105]	CCA tissues	Upregulated	miR-548b-3p	Various genes predicted

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
