# Peer review of "Current Advances in Basic and Translational Research of Cholangiocarcinoma"

_cancers, 2021, doi:10.3390/cancers13133307_

Round 1

Reviewer 1 Report

Dr Sato and colleagues presenta review article on basic and translational research in cholangiocarcinoma. The issue is of interest and the authors are key opinion leaders in the field. The manuscript is complete and clearly highlights advances and limitations in the field.

I congratulate the authors for the nice work and do not have major comments

Author Response

We appreciate your comments.

Reviewer 2 Report

The review is well written and provides an extensive overview of the recent advances in Cholangiocarcinoma. The following is suggested for minor revision:

  1. Please include a table detailing the different genetic mice models for Cholangiocarcinoma.
  2. The gut microbiota is an emerging player in different cancers, including Cholangiocarcinoma. It would be a nice to include this information to the review.

Author Response

Reviewer #2

Comments from Reviewer #2

Please include a table detailing the different genetic mice models for Cholangiocarcinoma.

Our response

We added a table to list genetic mouse models as Table 1.

Comments from Reviewer #2

The gut microbiota is an emerging player in different cancers, including Cholangiocarcinoma. It would be a nice to include this information to the review.

Our response

We added a new section to discuss the association of gut microbiota with CCA.

Reviewer 3 Report

In this review Sato and colleagues provide a comprehensive overview of advances in basic and translational research of cholangiocarcinoma.

This review compiles recent studies as shown by the references used, clearly showing the challenges to overcome in the detection and treatment of cholangiocarcinoma.

They introduce the disease showing the different types of cholangiocarcinoma depending on their anatomical region, as well as the combination of HCC-CCA. These facts show the difficulty in the diagnosis as well as the treatment.

The authors also show the genetic diversity in this disease, the presence of different mutations and aberrations. They refer to different recent studies that use transcriptomic and genomic sequencing to learn more about how genetic and epigenetic factors are involved in the pathogenesis of CCA.

The recent increase in studies on CCA shows interest in addressing this disease and show the challenges to overcome.

The authors also highlight the need for basic research, showing the benefits and limitations of in vitro and in vivo models. This review shows the latest advances in basic research such as the use of 3d and 2d cultures as well as their use in pdx and syngeneic models.

One of the clearest points to address is the search for biomarkers that allow their early detection, this being the most limiting factor.

I think this study deserves to be published because it very clearly summarizes the latest advances in the study of cholangiocarcinoma. It is written very clearly showing the benefits and limitations of the models used, being useful for future studies that seek to improve these models.

Author Response

We appreciate your comments.